Secreted Frizzled-related protein 4 inhibits the regeneration of hair follicles

Guo Haiying 1
Xing Yizhan 1
Deng Fang 1
Yang Ke 2
http://orcid.org/0000-0003-1684-207X Li Yuhong 1 liyuhongtmmu@hotmail.com
1 Department of Cell Biology, Army Medical University , Chongqing , China
2 Chongqing Stem Cell Therapy Engineering Technical Center, Children’s Hospital of Chongqing Medical University , Chongqing , China
Tobin Desmond
Electronic publication date: 2019 Jan 4
Publication date: 2019
Volume: 6
Electronic Location ID: e6153
Received 2018 Aug 28; Accepted 2018 Nov 24
Copyright: © 2019 Guo et al.
Copyright year: 2019
Copyright holder: Guo et al.
License: This is an open access article distributed under the terms of the Creative Commons Attribution License, which permits unrestricted use, distribution, reproduction and adaptation in any medium and for any purpose provided that it is properly attributed. For attribution, the original author(s), title, publication source (PeerJ) and either DOI or URL of the article must be cited.
License URL: https://creativecommons.org/licenses/by/4.0/

Keywords: Hair follicle, sFRP4, Wnt signaling pathway, Keratin

Funding: National Natural Science Foundation of China 81472895, 81872543 Municipal Natural Science Foundation of Chongqing cstc2018jcyjAX0053 This work was supported by the National Natural Science Foundation of China (No. 81472895, 81872543) and the Municipal Natural Science Foundation of Chongqing (No. cstc2018jcyjAX0053) The funders had no role in study design, data collection and analysis, decision to publish, or preparation of the manuscript.

==============================
Secreted Frizzled-related Protein 4 (sFRP4) belongs to Wnt inhibitors. Previously, we reported that sFRP4 inhibited the differentiation of melanocyte. Here, by using of immunostaining, we showed that sFRP4 is expressed in both human and mouse hair follicles, especially in the outer root sheath and inner root sheath. To reveal the role of sFRP4 in hair follicle growth and hair cycle, we induced synchronized hair cycle in the dorsal skin of mice by depilation, and injected sFRP4 intradermally into the skin. By hematoxylin and eosin staining, we found that the regeneration of hair follicles was inhibited by sFRP4. However, the structure of hair follicles remained complete. Compared with phosphate buffer saline-treated hair follicles, the sFRP4-treated hair follicles still had the same expression pattern of keratins. Our findings reveal that sFRP4 inhibits but not blocks the regeneration of hair follicles, and supply a potential therapeutic application to treat hair follicle regeneration disorders.

Introduction

The Wnt proteins play essential roles in the development, proliferation, differentiation, and migration of various cell types. Wnts transduce signals by binding to Frizzled receptor family and low-density lipoprotein-related protein 5/6 (LRP5/6) co-receptors. Activation of Wnt signaling is also controlled by the extracellular antagonists that can be divided into two classes, the secreted Frizzled-related proteins (sFRP) class and the Dickkopf (DKK) class. The sFRP class includes sFRP1, sFRP2, sFRP3, sFRP4, sFRP5, Wnt inhibitory factor 1, and Cerberus. They antagonize Wnt signaling by interacting with Wnt proteins directly. The DKK class includes DKK1, DKK2, DKK3, DKK4. They antagonize Wnt signaling by binding to LRP5/6 co-receptor complex.

Secreted Frizzled-related Protein 4 is a member of the sFRP family and has been reported to play roles in cell proliferation, differentiation, apoptosis, and carcinogenesis (Drake et al., 2009; Ford et al., 2013; Huang et al., 2010; Maganga et al., 2008; Park et al., 2008). Recently, several studies have demonstrated the importance of sFRP4 in skin tissue (Bai et al., 2015; Bayle et al., 2008; Chen et al., 2014; Maganga et al., 2008). In lesional skin of psoriasis patients and mouse models, the expression of sFRP4 were significantly decreased at both mRNA level and protein level. On the contrast, the expression level of sFRP4 protein was increased in the skin of the systemic sclerosis patients and tight-skin mouse compared to healthy skin (Bai et al., 2015; Bayle et al., 2008). In cultured human keratinocytes, exogenous sFRP4 inhibited the proliferation of keratinocyte (Bai et al., 2015; Maganga et al., 2008). Furthermore, sFRP4 also promotes the apoptosis and differentiation of human keratinocytes in vitro (Maganga et al., 2008). Previous studies also reported that sFRP4 functioned as an extra-follicular modulator, and coordinated the hair follicle cycling behavior (Chen & Chuong, 2012; Chen et al., 2014; Plikus et al., 2011).

Previously, we have investigated the role of sFRP4 in melanogenesis. We observed that sFRP4 was preferentially expressed in keratinocytes adjacent to melanocytes in epidermis. However, the role of sFRP4 in hair follicle regeneration has not been well elucidated. To address this issue, we determined the expression pattern of sFRP4 in the skin of human and mouse, treated the mouse skin with sFRP4 protein, and observed the treated skin and hair follicles systematically. We found that sFRP4 inhibited the regeneration of hair follicles.

Materials and Methods

Animals and skin samples

C57 BL/6 mice were obtained from and housed in the laboratory animal center of the Army Medical University. All the animal-related procedures were in strict accordance with the approved institutional animal care and maintenance protocols. All experimental protocols were approved by the Laboratory Animal Welfare and Ethics Committee of the Third Military Medical University. Permission number for producing animals: SCXK (YU)-20170002. Permission number for using animals: SYXK (YU)-20170002.

Human skin samples were obtained from 30 to 40 years old donors in the second affiliated Hospital of the Army Medical University. The use of the samples was consented verbally by the donors. The research was approved by the ethics committee of the Third Military Medical University.

Immunofluorescence

Immunostaining was performed on five μm sections from tissue samples. Sections were dewaxed, rehydrated, and boiled in citrate buffer solution. Then the sections were blocked with 10% goat serum, incubated with the following primary antibodies: goat anti-sFRP4 (1:100; Abcam, Shanghai, China), rabbit anti-K6 (1:50; Sangon Biotech, Songjiang, China), rabbit anti-K10 (1:50; Sangon Biotech, Songjiang, China), rabbit anti-K14 (1:50, Sangon Biotech, Songjiang, China), and rabbit anti-K19 (1:50, Sangon Biotech, Songjiang, China), and rabbit anti-β-catenin (1:200; Abcam, Cambridge, MA, USA). After washing, the sections were incubated with Alexa Fluor 488 (Invitrogen, Carlsbad, CA, USA) or Cy3 (Beyotime, Haimen, China) labeled secondary antibodies. After that, sections were counterstained with 4′,6-diamidino-2-phenylindole (DAPI) for nuclei visualization. At last, the cover slides were moved to microscope slides, mounted with antifade mounting medium (Beyotime, Haimen, China), and observed under fluorescent microscope.

Intradermal administration

To induce synchronized hair cycle, the hairs on the dorsal skin of 7-week-old C57 mice were depilated as described previously (Muller-Rover et al., 2001). A total of 25 μL recombinant sFRP4 (50 μg/mL, R&D Systems, Minneapolis, MN, USA) or phosphate buffer saline (PBS) (control) was administered intradermally into the dorsal skin after depilation. At 2, 3, or 4 days after the administration of sFRP4, the dorsal skin samples were harvested and prepared for analysis of histology or immunostaining.

Hematoxylin and eosin staining

The skin collected from sFRP4 or PBS administrated region were fixed with 4% paraformaldehyde overnight, gradually dehydrated in graded alcohol solutions, cleared in xylene, and subsequently embedded in paraffin. Sectioned samples were dewaxed, rehydrated, stained with hematoxylin (ZSGB-bio, Beijing, China) for 2 min and subsequently rinsed with water. The sections were later stained with eosin (ZSGB-bio, Beijing, China) for 2 min and rinsed with water thereafter. At last, the sections were dehydrated gradually, mounted with neutral gum (ZSGB-bio, Beijing, China) and observed under a microscope.

Statistical analysis

Data were presented as a mean ± SD of three independent experiments. Statistical analyses were evaluated using t-test and P < 0.05 was considered statistically significant.

Results

sFRP4 is expressed in human and mouse hair follicles

At first, we detected the endogenous expression of sFRP4 in human scalp skin by immunofluorescence. In epidermis, sFRP4 was expressed in all the layers (Figs. 1A–1C). In the human hair follicle, sFRP4 was strongly expressed in the inner root sheath (IRS) and the outer root sheath (ORS), weaker in the hair shaft (HS) and pre-cortex (Figs. 1D–1I).

Figure 1 The expression of sFRP4 in human scalp.

(A–C) The expression of sFRP4 in epidermis. (D–F) The expression of sFRP4 in the bulge area of hair follicles. (G–I) The expression of sFRP4 in the hair bulb area of hair follicles. The nuclei were counterstained with DAPI. C, F and I represent the merger of A and B, D and E, and G and H, respectively. Ep, epidermis; HS, hair stem; ORS, outer root sheath; IRS, inner root sheath; Co, precortex; DP, dermal papilla; Scale bar = 50 μm.

Then we detected the expression pattern of sFRP4 in mouse dorsal skin. Among all the hair cycle stages, sFRP4 was widely expressed in the hair follicle (Figs. 2A–2I). In anagen, nearly all the hair follicle structures can be observed. Double-labeling immunostaining further confirmed that sFRP4 was expressed at the ORS (K14 positive) and IRS (K10 positive) region in anagen hair follicles. The expression of sFRP4 was also detected in the IRS precursors and hair matrix region of mouse hair follicles (Figs. 2J–2Q).

Figure 2 The expression of sFRP4 in mouse dorsal skin.

(A–I) The expression of sFRP4 (red color) in hair cycle. (A–C) Postnatal day 8, anagen. (D–F) Postnatal day 16, catagen. (G–I) Postnatal day 21, telogen. (J–M) The expression of sFRP4 (red color) and K14 (green color) in anagen mouse skin. (N–Q) The expression of sFRP4 (red color) and K10 (green color) in anagen mouse skin. C, F, I, M and Q represent the merging of A and B, D and E, G and H, J–L and N–P, respectively. The nuclei were counterstained with DAPI. Scale bar = 50 μm.

sFRP4 inhibits the growth of hair follicle in vivo

The expression pattern of sFRP4 in hair follicle suggests that sFRP4 may play some roles in hair follicle cycle. To find out the roles of sFRP4 in hair follicle cycle and hair follicle regeneration, we induced synchronized hair growth by depilation and injected sFRP4 protein intradermally into the mouse dorsal skin (Fig. 3A). At 2 days after injection, the regeneration of the hair follicle was inhibited. However, the regeneration of the hair follicle was not blocked, it went on growing after the injection of sFRP4 stopped (Figs. 3C–3J). We also measured the length of hair follicles in the pictures of hematoxylin and eosin (H&E) staining with Image Pro Plus 6.0 software. Statistical analysis revealed that hair follicles in the sFRP4-injected group showed decreased hair length at 4 days after depilation (Fig. 3B).

Figure 3 sFRP4 inhibited the growth of hair follicle.

(A) The working model for the animal experiment. The hairs of 7-week-old mice were depilated, and sFRP4 was injected. The samples were obtained at day 2 and day 4 post injection. (B) The length of regenerated hair follicles in PBS-treated or sFRP4-treated mouse skin. N = 3, *P < 0.05. (C–J) H&E staining displays the structure of hair follicles in the PBS-treated or sFRP4-treated mouse skin. D, F, H and J are the enlargements of the framed area in C, E, G and I, respectively. 2d-PBS, 4d-PBS, 2 or 4 days after the injection of PBS. 2d-sFRP4, 4d-sFRP4, 2 or 4 days after the injection of sFRP4. Scale bar = 100 μm.

sFRP4 does not affect the expression pattern of structure markers in hair follicles

K6, K10, K14, and K19 were expressed in the regenerated hair follicles, both in the PBS-treated group and sFRP4-treated group. K6 and K14 were expressed in both IRS and ORS. K10 was expressed in the IRS. K19 was expressed in the bulge region (Fig. 4). The expression patterns were similar between the two groups, but the regeneration of hair follicles in sFRP4-treated group was delayed.

Figure 4 The expression of keratins in regenerated hair follicle.

The expression of keratins K6 (A and E), K10 (B and F), K14 (C and G), K19 (D and H) were determined by immunofluorescence. (A–D) The expression of keratins in sFRP4-injected samples. (E–H) The expression of keratins in PBS-injected samples. The nuclei were counterstained with DAPI. Scale bar = 50 μm.

sFRP4 inhibits the nuclear translocation of β-catenin in hair follicles

Beta-catenin expressed in the regenerated hair follicles, both in the PBS-treated group and sFRP4-treated group. In PBS-treated group, β-catenin was expressed in both nucleus and cell plasma (Figs. 5A and 5C). At 2 days after treatment, only few nuclear translocation of β-catenin was observed (Fig. 5A). At 4 days after treatment, more nuclear translocation of β-catenin was observed (Fig. 5C). In sFRP4-treated group, β-catenin was expressed in cell plasma (Figs. 5B and 5D).

Figure 5 The expression of β-catenin in regenerated hair follicle.

The expression of β-catenin was determined by immunofluorescence. (A) and (B) The expression of β-catenin at 2 days after treatment. (C) and (D) The expression of β-catenin at 4 days after treatment. (A) and (C) show the expression of β-catenin in PBS-injected samples. (B) and (D) show the expression of β-catenin in sFRP4-injected samples. The nuclei were counterstained with DAPI. Scale bar = 10 μm.

Discussion

Hair follicle grows periodically. The hair cycle is consist of anagen, catagen, and telogen. The regeneration of hair follicles begins with anagen. It is regulated by many factors, including factors from macroenvironment such as the level of hormone and mood, and factors from microenvironment such as adjacent cells. Secreted factors impact the regeneration of hair follicles through autocrine, paracrine, or endocrine. sFRP4 is a member of sFRP. It is reported that sFRP4 took effects via autocrine or paracrine (Bafico et al., 1999). sFRP4 is expressed in various tissues normally including endometrial stroma, pancreas, stomach, colon, lung, skeletal muscle, testis, ovary, kidney, heart, brain, mammary gland, cervix, eye, bone, prostate, and liver (Pawar & Rao, 2018). In the hair follicle, we found that sFRP4 was expressed in nearly all the keratinocytes, including ORS, IRS, matrix, and epidermis. As to hair cycle, sFRP4 was expressed in all the stages, including anagen, catagen, and telogen, which is in accordance with previous report (Chen et al., 2014). This expression pattern suggests that sFRP4 may play some roles in all the layers of hair follicle and epidermis. The expression pattern of sFRP4 in human scalp was similar to the expression pattern of sFRP4 in mouse dorsal skin. This suggests that sFRP4 may play similar role in different species.

After depilation, hair follicles enter anagen immediately (Muller-Rover et al., 2001). When sFRP4 was injected into the mouse skin, the regeneration of hair follicle was inhibited. The structures of ORS, IRS, and matrix were smaller than the control, and the length of hair follicle was shorter. However, the hair follicle entered anagen as well. This suggests that sFRP4 may not block the anagen onset, but inhibit the structure formation of hair follicle. Previously, we reported that sFRP4 inhibited the differentiation of melanocytes (Guo et al., 2017). Here, sFRP4 may play its role via inhibiting the proliferation and differentiation of precortex cells, transit amplifying cells, and hair follicle stem cells.

Usually, sFRPs take effects by binding with Wnt ligands, thus blocking the interaction of Wnt ligands and Wnt receptors. However, sFRPs can also augment Wnt signaling under certain conditions (Kawano & Kypta, 2003). Canonical Wnt ligands such as Wnt3a and Wnt10b can induce hair follicle regeneration (Li et al., 2013). Non-canonical Wnt ligands such as Wnt5a could inhibit the anagen onset of hair cycle (Xing et al., 2016). Translocation of β-catenin from plasma to nucleus is a key point for Wnt signaling pathway. Blocking of the translocation of β-catenin inhibited the anagen onset of hair regeneration. Here, sFRP4 inhibited the growth of hair follicle. It also inhibited the nuclear translocation of β-catenin. It is similar to the situation of the inhibition of canonical Wnt signaling pathway. Thus, sFRP4 may take effects by binding with Wnt ligands.

Conclusions

From the results of the present study, we conclude that sFRP4 is expressed in both mouse and human hair follicles. During anagen, overexpression of sFRP4 inhibits the growth of hair follicles through inhibiting the nuclear translocation of β-catenin, but would not block the growth of hair follicles. The hair follicles regenerated from sFRP4 treated skin have complete hair follicle structure.

We thank Prof. Jin Yang for her constructive suggestions during the experiments.

Additional Information and Declarations

Competing Interests

Author Contributions

Human Ethics

Animal Ethics

Data Availability

The authors declare that they have no competing interests.

Haiying Guo performed the experiments, prepared figures and/or tables, authored or reviewed drafts of the paper, approved the final draft.

Yizhan Xing performed the experiments, analyzed the data, approved the final draft.

Fang Deng performed the experiments, approved the final draft.

Ke Yang performed the experiments, contributed reagents/materials/analysis tools, approved the final draft.

Yuhong Li conceived and designed the experiments, prepared figures and/or tables, authored or reviewed drafts of the paper, approved the final draft.

The following information was supplied relating to ethical approvals (i.e., approving body and any reference numbers):

The Ethics Committee of The Third Military Medical University granted ethical approval to carry out the study within its facilities.

The following information was supplied relating to ethical approvals (i.e., approving body and any reference numbers):

All experimental protocols were approved by the Laboratory Animal Welfare and Ethics Committee of the Third Military Medical University. Permission number for producing animals: SCXK (YU)-20170002. Permission number for using animals: SYXK (YU)-20170002.

The following information was supplied regarding data availability:

Li, Yuhong (2018): raw data for sFRP4 and hair follicle-R1.rar. figshare. Figure. https://doi.org/10.6084/m9.figshare.7296686.v1.

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
