# Peer review of "Secreted Frizzled-related protein 4 inhibits the regeneration of hair follicles"

_PeerJ, doi:10.7717/peerj.6153_

## Round 0.1 · original submission · Major Revisions

The reviewers and Editor are interested in this work, and so would be willing to consider a suitably revised manuscript based on the reviewers' comments and suggestions for improvements.

Reviewer 1 ·

Basic reporting

no comment

Experimental design

no comment

Validity of the findings

no comment

Additional comments

In this manuscript, Guo at al. demonstrate an inhibitory effect of sFRP4 on hair growth. Although this result is interesting, I have issues with insufficient and superficial data analysis and interpretation.
1. This manuscript could be sufficiently improved if authors demonstrate that the effect of sFRP4 on hair growth is mediated via inhibition of Wnt pathway. Expression of cytoplasmic and nuclear beta-catenin after sFPRP4 injections would provide a good evidence of activation/inhibition of the Wnt signalling.
2. Authors should describe more clearly the effect of sFRP4 on hair growth. Quantifications of hair cycle stages as per Muller-Rover at al. (2001) would help to clarify the effect of sFRP4 on anagen initiation and progression.
3. In the results section, the authors state that sFRP4 does not affect the structure of hair follicles (lines 109-113). However, in the discussion section, the author’s claim that “The structures of ORS, IRS and matrix were smaller than the control” (lines 127-128). Furthermore, “sFRP4 may … inhibit the structure formation of hair follicle” (lines 129-130). These parts of the results and discussion sections need improving and clarification.

Reviewer 2 ·

Basic reporting

No comment.

Experimental design

No comment.

Validity of the findings

No comment.

Additional comments

In this study, the authors want to identify the role of Sfrp-4 in hair regeneration cycle. Through observing the expression pattern of Sfrp-4 and functional assay, they found that Sfrp-4 can inhibit but not block the regeneration cycle. Although the topic is important, there are several issues that the authors should take care and address to confirm their hypothesis.

1. Previous study had demonstrated that Sfrp-4 was expressed during the refractory telogen period (Chen et al., 2014). However, in this study, the author only showed the expression pattern during the anagen period. To evaluate the expression pattern of Sfrp-4 during the whole hair regeneration cycle period, including anagen and telogen is recommended.
2. If the Sfrp-4 did inhibit the hair regeneration, why is it expressed strongly in ORS and IRS during anagen period? According to this, the expression of Sfrp-4 should express strongly in telegen period which inhibit the hair stem cell entering anagen period.
3. In figure 3, the author measure the length of regenerated hair follicles to prove that Sfrp-4 did inhibit the regeneration of hair follicle. Can the author explain when they measure the length? Day 4 or the end of the anagen? To measure the full anagen hair follicle length is recommended to determine if Sfrp-4 diminish the period of anangen which let the hair follicle length become shorter.



Chen CC, Murray PJ, Jiang TX, Plikus MV, Chang YT, Lee OK, Widelitz RB, Chuong CM. Regenerative Hair Waves in Aging Mice and Extra-Follicular Modulators Follistatin, Dkk1, and Sfrp4. J Invest Dermatol 2014; 134(8): 2086-96.

---

## Round 0.2 · Minor Revisions

Thanks you for your careful attention to the reviewers' comments, and for the additional data, and the extended analysis of the overall data. I would ask the authors to label all histology/IHC images to allow for greater access of the reader to the data via more informative legends. I suggest each legend to be more informative of the result shown.

---

## Round 0.3 · accepted · Accept

Thank you for addressing these last minor issues.

#